# Differences in housing wealth between U.S. military service personnel and the Civilian population—Exploring the role of financial stress

Eric Olsen, Cäzilia Loibl *, Sherman D. Hanna, Andrew Hanks

Department of Human Sciences, The Ohio State University, Columbus, Ohio, United States of America

* loibl.3@osu.edu

## Abstract

The study investigates whether frequent job-based relocations, which are typical for military service personnel, are correlated with households' ability to accumulate housing wealth. Specifically, we investigate whether differences in homeownership rates and home equity values exist for two cohorts of military personnel, the older Korea/Vietnam and the younger post-Vietnam cohorts of servicemembers. The study accounts for individual financial stress and expectations about the economy, and controls for a rich set of demographic and socio-economic factors. Data sources are the 2022 Survey of Consumer Finances and the 2021 National Financial Capability Study. Results show that the two cohorts of military households do not differ from civilian households with regard to the home ownership rate. Greater individual financial stress on one hand and more positive economic expectations on the other hand emerge as two opposing, but stress-related factors linked to lower homeownership rates. The Post-Vietnam military personnel cohort is associated with lower home equity values compared to civilian households, but no difference was found between the Korea/Vietnam cohort and civilian households. From a policy perspective, our findings indicate that housing-focused efforts in the military, such as targeted loan products, relocation allowances, financial education and counseling programs, appear to help military households cope with the demands of military career paths and the transition to post-active life. When limiting the sample to homeowners, the data indicate lower housing wealth accumulation among the younger, Post-Vietnam era military households, compared to civilian households. As frequent military moves may prevent these households from building housing wealth while in the service, this group has had less time to accumulate housing wealth, documenting the role of housing tenure length for wealth accumulation for this unique population group.

**Data availability statement:** Data cannot be shared publicly because the data used in the study include publicly-restricted variables. Data are available from the FINRA Investor Education Foundation (contact Gary.Mottola@finra.org) for researchers who meet the criteria for access to confidential data.

**Funding:** The first author would like to thank the Air Force Institute of Technology for generous financial support.

**Competing interests:** The authors have declared that no competing interests exist.

## Introduction

Housing wealth is one of the most important assets for the economic security of individuals and families, across time, wealth levels, education levels, and ethnic/racial groups [1]. Home ownership provides unique benefits in the accumulation of wealth as forced savings and [2–5], as a protective buffer to weather economic shocks [6,7] and material hardship [8,9]. From a behavioral perspective, homes can also act as a consumption good [5]. Due to the critical role of housing wealth for the economic security of individuals and families across the life span [6], understanding of why certain population groups may lack behind in accumulating housing wealth is important information when analyzing the economic security of individuals and families.

A key hindrance in building housing wealth is domestic migration, whether due to professional or personal circumstances. Moving to a different location typically involves transition to new housing and the costs incurred with buying and selling a home [10], and possibly indirect costs of transitioning a family to a new location, especially lost income due to family members' un/and underemployment [11]. This study investigates the role of domestic migration from the perspective of current and former servicemembers of the United States military, referred to in this paper as "military service personnel". This population, currently about 6% of the adult population in the U.S. [12], provides a suitable background for studying the challenges of building housing wealth. The military population is required to relocate every two to three years while in active duty [13,14]. In addition, detailed data are available on the military population, which facilitates empirical research. We investigate housing wealth of military service personnel for two distinct cohorts. An older cohort who served during the Vietnam/Korean war era, and a younger cohort who served in the post-Vietnam era, in order to account for the role of time in accumulating housing wealth and the role of military cohort for the stress level in military personnels' lives. Specifically, this study asks, does the older, Korea/Vietnam era cohort of current and former military service personnel achieve housing wealth that is similar to civilian households? Does the younger, post-Vietnam era cohort of current and former military service personnel report lower housing wealth compared to civilian households? A particular focus in this context is on the link between financial stress and housing wealth, examining both individual financial stress and expectations about the economy.

### Economic frameworks for housing wealth accumulation

From an economic perspective, housing wealth formation can be explained through the Behavioral Life Cycle (BLC) framework proposed by Shefrin and Thaler [15]. The original Life-Cycle Hypothesis of Savings by Modigliani and Brumberg [16], and then later advanced by Ando and Modigliani [17], aims to explain household decisions regarding savings and consuming, taking into account household wealth as well as income. Under the Life-Cycle Hypothesis, rational households strive for intertemporal consumption smoothing by dissaving early in life to develop human capital. As earnings typically exceed expenses later in their work life, households build wealth to ultimately enter a period of dissaving in retirement. The Behavioral Life Cycle

framework incorporates psychological factors into this framework to account for human decision processes over time. From this perspective, the demands of military careers and the transition to post-active life can create economic shocks and delay housing wealth accumulation.

The Financial Capability framework [18,19] adds the perspective on specific financial factors that can influence economic decisions. It posits that financial capability is derived from both internal *ability to act* and external *opportunity to act.* Internal factors, such as financial knowledge, skill, intelligence, and behaviors, comprise the *ability* component of financial capability, while external factors such as access to quality financial products, financial professionals, and policies comprise the *opportunity* component [18,19]. This framework is applicable to the investigation of housing wealth for US military service personnel because they are subject to housing-specific internal and external factors. They have access to housing-specific financial education resources while on active-duty and certified housing counseling and educational resources to increase financial knowledge and management skills during and after military service [20]. The Veteran's Administration provides military service personnel access to unique relocation allowances, mortgage products with no down payment requirement, and lower closing costs, thereby removing one of the largest external hurdles for new home buyers, the liquidity requirement [21,22].

The transitory aspect of military careers presents economic shocks unique to service members. The Behavioral Life Cycle framework suggests that consequently these individuals may delay housing wealth accumulation. In contrast, the financial capability framework suggests that military-specific support services and financial products can support housing wealth accumulation at a rate equal to the civilian population. These opposing perspectives point to a gap in available research.

## Differences in military service personnel by enlistment and era of military service

Military service personnel can be grouped into cohorts, with common approaches being enlistment process (draft or voluntary) and by era of military service (e.g., pre-Vietnam Era (prior to 1964), Vietnam Era (1964–1973), post-Vietnam Era (after 1973)) [23,24]. World War II, The Korean War, and The Vietnam War instituted a mandatory draft lottery of U.S. civilian males, while post-Vietnam conflicts solely engaged volunteer soldiers. For example, U.S. soldiers in the Vietnam War were about one third drafted soldiers and two-third volunteer soldiers, although it is estimated that roughly half of the volunteer soldiers enlisted to avoid being drafted in the future [25]. As a result, the Korea/Vietnam and earlier military service personnel present a better representation of the general population than post-Vietnam military service personnel. With a focus on housing wealth, this older cohort, which is age 67 and older as of the 2022 Survey of Consumer Finances (the cutoff is age 18 by 1973, the last year of the draft), had a longer time to accumulate housing wealth. In contrast, post-Vietnam era military personnel, who enlisted voluntarily and were between the ages of 18 and 66 as of the 2022 Survey of Consumer Finances, present a more unique population that had less time to accumulate housing wealth. We acknowledge that this approach to defining military service cohorts has two shortcomings. First, the approach does not correctly categorize individuals who were military age in 1973 but did not join the military until after the Vietnam war; these are included in the Vietnam era cohort in the current study. Second, Post-Vietnam military service personnel represents a particularly broad range of military experiences and stressors. Those who have served during and after the Global War on Terrorism following the 2001 attacks to the United States have seen higher levels of combat, deployments, and stress than those who served between the Vietnam War and the Global War on Terrorism, from 1973 to 2001, as documented, for example, with the Millenium Cohort Study [26,27]. The greater exposure to combat has been connected to higher likelihood of having a service-connected disability when compared to Vietnam Era and earlier military service personnel [23]. The post-2001 military experience may also explain difficulties in re-integration, higher drug use, unemployment, and the experience of stress among military personnel serving during and after the Global War on Terrorism [25,28]. Due to sample size limitations, the current study focuses on the two larger cohorts, prior to/during vs after the Vietnam Wars, to respond to a gap in current research but notes that future

research should aim to assess financial vulnerability of military services personnel by taking into account a finer-grained analysis of cohorts.

## Financial stress among military service personnel

U.S. military Veterans have been the subject of considerable academic research. Econometric research, such as by Angrist and Krueger [24], utilized the natural experiment of draft selection to highlight the econometric power of instrumental variables, and found that service in World War II was related to lower Veterans post-service earnings relative to not having served. A growing number of studies examine the elevated levels of stress as an outcome of the military environment, such as regular required relocations, deployments, combat, injuries, service-related disabilities, and the role of stress for health, employment, social relationships, and economic security in the lives of military service personnel [29]. A meta-analysis of 33 studies of Veterans of Operation Enduring Freedom and Iraqi Freedom (OEF/OIF) found that 23% of these Veterans experienced posttraumatic stress disorder (PTSD) [30]. Mental illnesses, such as PTSD and traumatic brain injury (TBI), have been shown to increase the risk of suicidal behavior in Veterans [31]. For OEF/OIF Veterans, rates of suicide have outpaced those of civilians and in some months and years the total number of suicides has surpassed combat deaths [32,33]. Another line of research has linked stress to significantly elevated rates of homelessness among Veterans relative to the general population [34].

Among the dimensions of stress for military service personnel, financial stress has received some attention. With regard to active-duty military service personnel, Department of Defense surveys rated finances as one of the most significant stressors, greater even than deployments and personal relationships [35]. A qualitative study at seven installations across all four military branches found financial management issues to be the most commonly cited personnel problem [36]. A study of 701 U.S. Army soldiers shortly before a year-long deployment to an active war zone found that higher anxiety was the most important predictor of concerning financial behaviors [37].

Turning to individuals who separated from military service, a study of recently separated military service personnel from 0 months to 36 months post-service found about one in eight post-service Veterans felt poorly equipped to manage their personal finances [29]. The study specifically examined housing stability and found a high correlation between experiencing housing instability and having a vulnerable financial status. When examining post-service personnel across socio-demographic groups, an industry report based on data from the 2015 National Financial Capability Study documents, in descriptive analyses, that an overall measure of financial standing was slightly better for Veterans than civilians. Veterans had better savings and investment outcomes and experienced slightly better financial conditions relative to civilians [38]. Taken together, the studies indicate the possibility of financial stress during active service and immediately after discharge. Over time, improved financial outcomes have been observed. Less attention, however, has been focused on housing wealth accumulation among current and former military service personnel. Our working hypothesis in response to this gap in research is that the older, Korea/Vietnam era cohort of military service personnel accumulates housing wealth similar to the civilian population, if we control for all plausible characteristics available in our dataset. The younger, post-Vietnam era cohort may still show disparities in housing wealth, partly because of less time after service.

## Association of financial stress and housing wealth

A number of studies have examined the relationship between financial stress and housing wealth accumulation. Financial stress has been defined in a number of ways, including objective measures, such as debt-to-income ratio, psychological measures, such as anxiety, and measures of the ability to cope, such as low financial literacy [39].

Using financial stress as an outcome measure, data from the 2018 National Financial Capability Study show that the presence of a home equity loan, a mortgage default, and the occurrence of negative home equity is associated with greater financial stress across four different measures [40]. Similarly, a study using the 2018 National Financial Capability Study showed that not only is mortgage delinquency positively associated with financial anxiety, but the effects are even

stronger when interacted with a variable measuring financial capability. Among those who had experienced a mortgage delinquency, individuals with higher financial capability (measured as a composite of objective and subjective financial knowledge, perceived capability, and desired financial behavior) experienced greater stress than individuals with lower financial capability [41]. Addressing the issue of reverse causality, a study using a quasi-experimental empirical strategy and data from the Health and Retirement Study found that the US housing boom from the mid-1990s through 2006 and subsequent housing appreciation, was associated with significantly lower risk of anxiety, especially for women. The authors conclude that an increase in housing wealth is interpreted by consumers as an increase in overall wealth, and that consumers are cognizant of how fluctuations in the housing market may be linked to their financial stability [42].

A separate stream of research documents the role of financial stress for disparities in housing wealth accumulation. Some studies have reported an association of financial vulnerability and the ability to purchase a home and accumulate home equity, often through a racial lens [43]. During the Great Recession and the COVID-19 pandemic, government programs were aimed at lowering financial stress to avoid financial hardship and the loss of the home. In conclusion, a direct link between housing wealth and financial stress has been reported, in both directions and based on a range of financial stress definitions.

In addition to financial stress experienced by households, expectations about the larger economic situation have been linked to housing wealth, pointing to the need for two types of stress measures. Individuals with more negative expectations about the greater economy have been shown to be more conservative in their consumption, credit, and investment behavior, while the opposite is true for individuals with optimistic expectations [44]. With regard to home equity, more uncertain expectations about the economy have been linked to lower likelihood of investing in risky assets, pointing to potential preferences for the stability of housing wealth, and to household behavior that shows more precaution in borrowing against the equity in the home, for example, with home equity lines of credit [45,46]. Our working hypothesis in response to this gap in research is that financial stress is associated with lower levels of home ownership and home equity. In contrast, we hypothesize that more negative expectations about the economy are associated with conservative financial behaviors, including a greater interest in investing in home purchase and home equity growth. Taken together, the current study investigates the following hypotheses:

*Hypothesis 1* The older, Korea/Vietnam era cohort of former military service personnel has similar rates of home ownership compared to civilian households.

*Hypothesis 2* The younger, Post-Vietnam era cohort of current and former military service personnel has lower rates of home ownership compared to civilian households.

*Hypothesis 3* The older, Korea/Vietnam era cohort of former military service personnel has similar home equity compared to civilian households, among those who own a home.

*Hypothesis 4* The younger, Post-Vietnam era cohort of current and former military service personnel has lower home equity compared to civilian households, among those who own a home.

In our empirical models, use a newly developed composite measure of individual financial stress, a measure of individuals' expectations about the economy, and control for a rich set of demographic and socio-economic factors.

## Data and methods

### Data

We use data from the 2022 Survey of Consumer Finances (SCF, N = 4,595) and from the 2021 National Financial Capability Study. The Survey of Consumer Finances is a triennial cross-sectional survey of U.S. families that includes information

on their balance sheets, income, demographics, objective financial literacy, behavioral attitudes, and use of financial products [47]. The survey is sponsored by the Federal Reserve Board in cooperation with the Department of the Treasury, with data collected at the University of Chicago.

This study contains two samples of the Survey of Consumer Finances. For H1 and H2, the samples include renters and homeowners (n = 4,531). For H3 and H4, the analysis is limited to households with home equity greater than $0 (n = 3,047). As the Survey of Consumer Finances provides multiple imputations in place of dropping observations with missing values, we utilize the J-coded shadow variables to verify confidence in the focal variables [48]. We removed 25 observations with J values for focal variables greater than 90 (0.5% of N). In addition, we removed respondents with negative or zero-dollar responses for household income (n = 39, 0.8% of N) and home equity (n = 15, 0.3% of N), to allow for log transformation of income and home equity values.

We use the 2021 National Financial Capability Study (N = 27,118) to develop a stress composite variable that measures the multiple facets of individual financial stress. The National Financial Capability Study is a US survey sponsored by the FINRA Investor Education Foundation and is nationally representative with about 500 respondents in all states except for California and Oregon. It has been administered on a triennial basis since 2009. This study is unique as it focuses on qualitative measures of financial capability, including questions that relate to financial anxiety, which are not included in the Survey of Consumer Finances.

## Measures

***Outcome measures:*** The two outcome variables, rate of home ownership and amount of home equity, are constructed through a series of questions inquiring about different types of asset ownership, market values of assets, and loan balances.

***Focal predictor – military status:*** Military status and cohort are established from the question, "(Have you/Has he/Has she/Has he or she) ever been in the military service? Include only service in U.S. military or national guard" with responses for the respondent or spouse or partner being combined into a variable for the Primary Economic Unit (coded as 1 for Yes and 0 for No service). We acknowledge that this question does not allow us to distinguish between active-duty and discharged or retired military personnel; this fact should be kept in mind when interpreting the results. Department of Defense civilian employees do not count as military personnel.

We divide the military households, which include the national guard, into two cohorts: Korea/Vietnam Era military service personnel (volunteer plus drafted soldiers) and Post-Vietnam Era military service personnel (all-volunteer soldiers). Respondents age 18 and older in 1973 (the final year of the Vietnam War draft) are coded as Korea/Vietnam Era military service personnel households, age 67 and older as of the time of the survey. Younger military service personnel households are coded as Post-Vietnam Era military service personnel households, younger than age 67 as of the time of the survey. In S1 Text, we provide the complete wording for all variables used to create the military status and financial stress measures.

***Predictor – financial stress:*** We use a measure of individual financial stress and a measure of financial stress related to expectations of the larger economy. For the measure of individual financial stress, a two-step approach is used that includes data from the National Financial Capability Study, see detailed information in S1 Table, S1 List, and to S2 Table. The individual financial stress composite variable ranges from 0.94 (low) to 7.28 (high).

For the measure of financial stress related to expectations about the economy, we use a question in the Survey of Consumer Finances, "I'd like to start this interview by asking you about your expectations for the future… Over the next year, do you expect the economy to perform better, worse, or about the same as now?". Responses are coded as 1 = Worse, 2 = About the same, and 3 = Better.

***Demographic and socio-economic control measures*** include indicators for age (18–95), sex (Male = 1), racial/ethnic status (White (omitted), Black, Hispanic, and Asian or other), Education (Less than high school degree or GED, High

school degree or GED, Some college or associates degree, and Bachelor's degree or higher (reference), marital status (Married or living with partner = 1), children (0 (reference), 1–2, 3–4, and 5+), work status (Work for other (reference), Self-employed/partnership, Retired/disabled/student/homemaker, Other not working (age <=64), and natural log of household income (6.07–19.94).

## Sample description

Table 1 displays statistics for the sample for homeowners and those who do not own a home. Homeowners comprise 68% of our sample. Korea/Vietnam Era and Post-Vietnam Era military service personnel households each comprise about 7% of the sample. There is a statistically significant difference in home ownership rates by military period (Korea/Vietnam Era) ($p < 0.001$). For the financial stress composite variable, homeowners have a lower mean at 4.12 vs. non-homeowners at 4.57 ($p < 0.001$), with statistically significant differences in all three sub-categories of 1-year economic expectations ($p < 0.001$). Sample descriptive statistics of Korea/Vietnam era households, Post-Vietnam era households, and civilian households are shown in S3 Table.

## Empirical approach

The Survey of Consumer Finances generates five implicates of each data record [48]. We use all five implicates with the repeated-imputation inference (RII) method in order to accurately represent the true variances that would be obtained with one implicate [49]. Data analyses include RII reduced-form binary logistic regression and an RII Ordinary Least Square (OLS) regression of home ownership (H1, H2) and level of home equity (H3, H4) respectively with the binary focal predictors of Korea/Vietnam Era military service personnel households and Post-Vietnam Era military service personnel households. The baseline specifications are as follows:

$$\ln\left(\frac{p\left(HM_i\right)}{1 - p\left(HM_i\right)}\right) = \alpha_0 + \alpha_1 KVMSP_i + \alpha_2 PVMSP_i \tag{1a}$$

$$\ln HM_i = \alpha_0 + \alpha_1 KVMSP_i + \alpha_2 PVMSP_i + \varepsilon_i \tag{1b}$$

With *HM* measuring our housing wealth dependent variable of individual i's household, either a binary measure of home ownership (logit HM) or the natural logarithm of home equity (log HM). *KVMSP* is a binary indicator of Korea/Vietnam Era military service personnel household status, and *PVMSP* is a binary indicator of Post-Vietnam Era military service personnel household status for individual i's household, with the alternative being civilian household. In Equation (1), $\varepsilon$ is a normally distributed random error component, and the coefficients of interest are $\alpha_1$ and $\alpha_2$.

We add two secondary predictors to Equation (1) to better understand the relationship between stress and housing wealth. The regression specifications are as follows:

$$\ln\left(\frac{p\left(HM_i\right)}{1 - p\left(HM_i\right)}\right) = \alpha_0 + \alpha_1 KVMSP_i + \alpha_2 PVMSP_i + \alpha_3 SC_i + \alpha_4 EE_i \tag{2a}$$

$$\ln HM_i = \alpha_0 + \alpha_1 KVMSP_i + \alpha_2 PVMSP_i + \alpha_3 SC_i + \alpha_4 EE_i + \varepsilon_i \tag{2b}$$

Here, *SC* is our stress composite variable that was imputed using shared variables between the National Financial Capability Study and the Survey of Consumer Finances, and *EE* is the one-year economic expectation, both for individual i. In Equation (2), the coefficients of interest are $\alpha_3$ and $\alpha_4$.

**Table 1. Sample Characteristics by Home Ownership.**

| Variables | (1) | | (2) | | (3) | | (4) |
|---|---|---|---|---|---|---|---|
| | Full sample | | Homeowner: | | Homeowner: | | T test |
| | | | Yes | | No | | |
| | Mean | (SD) | Mean | (SD) | Mean | (SD) | p-value |
| Homeowner (0/1) | 0.68 | (0.47) | – | – | – | – | – |
| Focal predictors | | | | | | | |
| Korea/Vietnam Era household (0/1) | 0.07 | (0.25) | 0.09 | (0.28) | 0.02 | (0.15) | 0.000 |
| Post-Vietnam Era household (0/1) | 0.07 | (0.25) | 0.07 | (0.25) | 0.06 | (0.24) | 0.057 |
| Stress variables | | | | | | | |
| Financial stress composite (0–9) | 4.26 | (0.72) | 4.12 | (0.65) | 4.57 | (0.77) | 0.000 |
| Economic expectations, 1-year | | | | | | | |
| Worse (0/1) | 0.48 | (0.50) | 0.52 | (0.50) | 0.40 | (0.49) | 0.000 |
| About the same (0/1) | 0.37 | (0.48) | 0.35 | (0.48) | 0.40 | (0.49) | 0.000 |
| Better (0/1) | 0.15 | (0.36) | 0.13 | (0.33) | 0.20 | (0.40) | 0.000 |
| Demographic and socio-economic controls | | | | | | | |
| Age (18–95) | 54.43 | (16.20) | 58.74 | (14.37) | 45.44 | (16.11) | 0.000 |
| Male (0/1) | 0.76 | (0.43) | 0.84 | (0.37) | 0.60 | (0.49) | 0.000 |
| Racial/ethnic status | | | | | | | |
| White (0/1) | 0.68 | (0.67) | 0.75 | (0.60) | 0.52 | (0.78) | 0.000 |
| Black (0/1) | 0.21 | (0.63) | 0.14 | (0.54) | 0.37 | (0.77) | 0.000 |
| Hispanic (0/1) | 0.19 | (0.62) | 0.14 | (0.54) | 0.30 | (0.75) | 0.000 |
| Asian & other (0/1) | 0.13 | (0.59) | 0.13 | (0.53) | 0.14 | (0.69) | 0.140 |
| Education | | | | | | | |
| Did not complete high school/GED (0/1) | 0.09 | (0.29) | 0.06 | (0.23) | 0.16 | (0.37) | 0.000 |
| High school graduate (0/1) | 0.20 | (0.40) | 0.16 | (0.37) | 0.27 | (0.44) | 0.000 |
| Some college or associates degree (0/1) | 0.22 | (0.41) | 0.19 | (0.39) | 0.27 | (0.44) | 0.000 |
| Bachelor's degree or higher (0/1) | 0.49 | (0.50) | 0.59 | (0.49) | 0.30 | (0.46) | 0.000 |
| Married or living with partner (0/1) | 0.63 | (0.48) | 0.75 | (0.44) | 0.39 | (0.49) | 0.000 |
| Children | | | | | | | |
| None (0/1) | 0.60 | (0.49) | 0.60 | (0.49) | 0.61 | (0.49) | 0.054 |
| 1-2 (0/1) | 0.32 | (0.46) | 0.33 | (0.47) | 0.29 | (0.45) | 0.000 |
| 3-4 (0/1) | 0.07 | (0.26) | 0.07 | (0.25) | 0.08 | (0.28) | 0.000 |
| 5+ (0/1) | 0.01 | (0.08) | 0.00 | (0.07) | 0.01 | (0.11) | 0.000 |
| Work status | | | | | | | |
| Work for other (0/1) | 0.50 | (0.50) | 0.45 | (0.50) | 0.61 | (0.49) | 0.000 |
| Self-employed or partnership (0/1) | 0.21 | (0.41) | 0.26 | (0.44) | 0.12 | (0.33) | 0.000 |
| Retired, disabled, student, homemaker (0/1) | 0.25 | (0.43) | 0.27 | (0.45) | 0.20 | (0.40) | 0.000 |
| Other not working, age<=64 (0/1) | 0.04 | (0.19) | 0.02 | (0.14) | 0.07 | (0.26) | 0.000 |
| Natural log of income (6.07–19.94) | 11.78 | (1.70) | 12.28 | (1.73) | 10.72 | (1.01) | 0.000 |
| N | 4,531 | | 3,062 (67.58%) | | 1,469 (32.42%) | | |

Lastly, we add demographic and socioeconomic controls to Equation (2) to account for observable and measurable differences between the three sub-populations of civilian households, Korea/Vietnam Era military service personnel households, and Post-Vietnam Era military service personnel households. The regression specifications are as follows:

$$\ln\left(\frac{p\left(HM_i\right)}{1 - p\left(HM_i\right)}\right) = \alpha_0 + \alpha_1 KVMSP_i + \alpha_2 PVMSP_i + \alpha_3 SC_i + \alpha_4 EE_i + \Gamma SES_i \tag{3a}$$

$$\ln HM_i = \alpha_0 + \alpha_1 KVMSP_i + \alpha_2 PVMSP_i + \alpha_3 SC_i + \alpha_4 EE_i + \Gamma SES_i + \varepsilon_i \tag{3b}$$

Here, *SES* represents the multiple demographic and socioeconomic control variables. In Equation (3), the coefficients of interest are $\alpha_1$, $\alpha_2$, $\alpha_3$ and $\alpha_4$, and according to our hypotheses, we expect the coefficients of $a_1$ through $a_4$ to be different than 0.

## Results

### Role of military status and cohort for home ownership

Model 1 is the binary logistic regression (Model 1 in Table 2) without controls for stress variables and socio-economic charac-teristics. When converting the coefficients shown in Table 2 to into odds-ratios by taking the exponential value of the unrounded coefficient of Korea/Vietnam Era, we obtain an odds ratio. The odds of Korea/Vietnam Era military service personnel house-holds owning a home is 4.23 times as large as the odds of civilian households owning a home ($p<0.001$). The association of military personnel status during the Post-Vietnam Era with home ownership was not statistically significant, at $p<0.05$.

Model 2 adds the stress variables to better understand their relationships to home ownership. Results show that for every unit increase in financial stress composite score, the odds of home ownership are 58.66% lower ($p<0.001$), how-ever, those who believe the economy to be better in the upcoming year had a 33.95% lower odds of home ownership as compared to those who believe the economy will be about the same ($p<0.001$), and those who thought the economy would be worse had a 45.46% higher odds of home ownership ($p<0.001$).

Model 3 tests H1 and H2 by including demographic and socio-economic control variables. In this model, our focal predictors of Korea/Vietnam Era and Post-Vietnam Era military service personnel households are no longer statistically significant, but the financial stress composite still holds a statistically significant relationship to home ownership with a 21.61% lower odds of home ownership with each unit increase in the stress composite score ($p<0.001$). Those who believe economic expectations will be better in one year had a smaller magnitude but still significant 27.19% lower odds of home ownership as compared to those who thought it would be about the same ($p<0.01$).

The statistically significant control variables were age, Black, Asian, or "other" relative to White race, those who were married, with children, and household income. Notably, Black respondents relative to White respondents had a 35.25% lower odds of home ownership ($p<0.001$), while Asian or "other" respondents had a 44.72% higher odds of home owner-ship when compared to White respondents ($p<0.01$). The combined effect of age and age-squared implies that on aver-age home equity increases with age, though at a decreasing rate. Married or living with partner respondents had a 2.20 higher odds of home ownership as compared to those who were not married or living with a partner ($p<0.001$), those with 1–2 or 3–4 children in the home had a 58.32% ($p<0.001$) and 46.79% ($p<0.05$) higher odds of home ownership respec-tively when compared to those without children ($p<0.001$). Finally, every unit increase in the natural log of income was associated with a 2.01 times higher odds of home ownership ($p<0.001$).

### Role of military status and cohort for home equity

Model 1 (Table 3) is an Ordinary Least Square regression without controls for stress variables and socio-economic characteristics. Korea/Vietnam Era military service personnel households had a 40.86% higher amount of home equity ($p<0.001$) whereas Post-Vietnam Era military service personnel households had a 38.73% lower amount of home equity ($p<0.001$) when compared to civilian households. As our dependent variable was the natural log of home equity, we expo-nentiate the coefficient to interpret the result as the percent change.

**Table 2. Results of Binary Logistic Regression of Home Ownership on Military Personnel Status Households, Stress, and Socio-economic Control Measures.**

| | Model 1 | Model 2 | Model 3 |
|---|---|---|---|
| | Coefficient (SE) | Coefficient (SE) | Coefficient (SE) |
| Focal predictors (ref. civilian) | | | |
| Korea/Vietnam Era | 1.44*** (0.19) | 0.93*** (0.19) | −0.04 (0.23) |
| Post-Vietnam Era | 0.19 (0.13) | 0.20 (0.14) | 0.19 (0.16) |
| Stress variables | | | |
| Financial Stress Composite | | −0.88*** (0.05) | −0.24*** (0.07) |
| 1-year Economic Expectations (ref. About the same) | | | |
| Worse | | 0.37*** (0.07) | 0.07 (0.09) |
| Better | | −0.41*** (0.10) | −0.32** (0.12) |
| Demographic and socioeconomic controls | | | |
| Age | | | 0.12*** (0.02) |
| Age Squared/10000 | | | −0.54** (0.17) |
| Male | | | −0.03 (0.12) |
| Race (ref. White) | | | |
| Black | | | −0.43*** (0.09) |
| Hispanic | | | −0.17 (0.10) |
| Asian or other | | | 0.37** (0.12) |
| Education (ref. Bachelor's degree or higher) | | | |
| Less than high school | | | −0.74*** (0.15) |
| High school | | | −0.25* (0.11) |
| Some college | | | −0.08 (0.11) |
| Married or living w/ Partner | | | 0.79*** (0.12) |
| Children (ref. none) | | | |
| 1-2 | | | 0.46*** (0.10) |
| 3-4 | | | 0.38* (0.16) |
| 5+ | | | −0.22 (0.42) |
| Work status (ref. work for others) | | | |
| Self-employed or partnership | | | −0.05 (0.13) |
| Retired, disabled, student, homemaker | | | 0.04 (0.13) |
| Other not working (age < 65) | | | 0.12 (0.21) |
| Natural log of income | | | 0.70*** (0.05) |
| Intercept | 0.65*** (0.03) | 4.40*** (0.23) | −10.88*** (0.83) |
| N | 4,531 | 4,531 | 4,531 |
| Log Likelihood | −14,067.06 | −13,028.80 | −9,450.90 |
| Likelihood Ratio Chi-Squared Test | 413.66*** | 2,490.19*** | 9,645.99*** |

* $p < 0.05$, ** $p < 0.01$, *** $p < 0.001$.

Model 2 adds the stress variables to better understand their relationships to home equity. Results show that being in the Korea/Vietnam Era military service personnel household category is no longer statically significant, however, being in the Post-Vietnam Era military service personnel household category is associated with a 37.22% lower home equity as compared to civilian households ($p < 0.001$). In addition, each unit increase in financial stress is associated with a 43.87%

**Table 3. Results of Ordinary Least Squares (OLS) Regression of Home Equity on Military Personnel Status Households, Stress, and Socio-economic Control Measures.**

| | Model 1 | Model 2 | Model 3 |
|---|---|---|---|
| | Coefficient (SE) | Coefficient (SE) | Coefficient (SE) |
| Focal predictors (ref. civilians) | | | |
| Korea/Vietnam Era | 0.34*** (0.09) | 0.06 (0.09) | −0.10 (0.06) |
| Post-Vietnam Era | −0.49*** (0.10) | −0.47*** (0.10) | −0.16* (0.08) |
| Stress variables | | | |
| Financial Stress Composite | | −0.58*** (0.04) | −0.03 (0.03) |
| 1-year Economic Expectations (ref. About the same) | | | |
| Worse | | 0.25*** (0.06) | −0.04 (0.04) |
| Better | | 0.001 (0.08) | 0.08 (0.06) |
| Demographic and socioeconomic controls | | | |
| Age | | | 0.06*** (0.01) |
| Age Squared/10000 | | | −0.36*** (0.07) |
| Male | | | 0.03 (0.07) |
| Racial/ethnic status (ref. White) | | | |
| Black | | | −0.43*** (0.05) |
| Hispanic | | | 0.01 (0.05) |
| Asian or other | | | 0.34*** (0.05) |
| Education (ref. Bachelor's degree or higher) | | | |
| Less than high school | | | −0.56*** (0.09) |
| High school | | | −0.38*** (0.05) |
| Some college | | | −0.30*** (0.05) |
| Married or living w/ Partner | | | 0.09 (0.06) |
| Children (ref. none) | | | |
| 1-2 | | | 0.08* (0.04) |
| 3-4 | | | 0.03 (0.07) |
| 5+ | | | 0.49 (0.29) |
| Work status (ref. work for others) | | | |
| Self-employed or partnership | | | 0.33*** (0.05) |
| Retired, disabled, student, homemaker | | | 0.28*** (0.05) |
| Other not working (age<65) | | | 0.40** (0.12) |
| Natural log of income | | | 0.50*** (0.02) |
| Intercept | 12.84*** (0.03) | 15.11*** (0.18) | 4.50*** (0.38) |
| N | 3,047 | 3,047 | 3,047 |
| F-test | 107.68*** | 283.09*** | 884.45*** |
| R-squared | 0.01 | 0.08 | 0.62 |

\* $p<0.05$, \*\* $p<0.01$, \*\*\* $p<0.001$.

lower home equity ($p<0.001$) and respondents with worse one-year economic expectations were associated with 28.25% higher home equity as compared to those who had a one-year economic expectations of about the same ($p<0.001$).

 Regarding H3 and H4, Model 3 again builds on the prior two models by including demographic and socio-economic control variables. In this model, our focal predictors of being a in the Korea/Vietnam era or Post-Vietnam era are not

statistically significant but being a Post-Vietnam Era military service personnel household is associated with a 14.41% decrease in home equity, when controlling for an array of demographic variables in the model ($p < 0.05$). The financial stress composite and 1-year economic expectation variables are not statistically significant.

The statistically significant and noteworthy control variables were again age, Black and Asian or other relative to White, 1–2 children relative to 0, and natural log of income. In addition, all three education categories relative to bachelor's degree or higher, and work status categories were significant for this model. Based on the combined effect of age and age squared, home equity increased over all ages in the sample. At the mean age, every unit increase in age was associated with an approximately 6% increase in home equity ($p < 0.001$). Black race was associated with 34.95% lower home equity ($p < 0.001$), and Asian race or other racial backgrounds were associated with 40.49% higher home equity ($p < 0.001$) when both were compared to White respondents. In addition, households with 1–2 children, relative to having none, were associated with an 8.81% increase in home equity ($p < 0.05$). At mean income, a one-unit increase in the natural log of income was associated with a 49.88% increase in home equity ($p < 0.001$). Relative to respondents with a bachelor's degree or higher, lower amounts of home equity were associated with lower educational attainment ($p < 0.001$). Self-employment and retirement status were associated with higher levels of home equity relative to being employed ($p < 0.01$).

## Discussion

This study contributes new insights into the intersections of housing wealth, military service, and financial stress. The study contributes new knowledge as one of the first studies that examine both home ownership and housing wealth for military service personnel broken down by era of military service and corresponding method of enlistment. An additional innovation is the use of a newly-developed measure of individual financial stress that is reflective of the multiple facets of both objective and subjective measures of financial stress.

Our main findings indicate, first, similar levels of housing wealth among military personnel and civilian households. Home ownership rates are similar to civilian households for both the older, Korea/Vietnam era miliary households and the younger, post-Vietnam era military households (H1 supported, H2 not supported). Second, when examining home equity among households with housing wealth, the younger, post-Vietnam military personnel cohort is associated with lower amounts of home equity than civilian households, as expected. The older Korea/Vietnam era miliary personnel cohort has accumulated similar amounts of home equity as civilian households, which was also hypothesized (H3, H4 supported).

The study further documents the role of individual financial stress, economic expectations, and demographic factors in the context of housing wealth. Greater individuals financial stress is found to be associated with lower homeownership rates, documenting that financial stress has multiple facets, as captured in this newly developed composite measure and that they relate to the home buying decision. The individual financial stress measure introduced in this study is derived from debt-to-income ratio, payday loan use, missed payments, credit card balance, lack of emergency savings, financial risk tolerance, and financial knowledge.

The results for economic expectations indicate that individuals with better one-year expectations are less likely to own a home compared to those that do not expect changes in the economy. This finding does not align with the initial working hypothesis that a more positive economic outlook is associated with homeownership, as compared to renting. The literature on association of economic expectations and household economic behavior documents that more positive economic expectations tend to make households take greater financial risks, such as through new borrowing and credit line increases [46]. Based on one-year frames of economic expectations, our findings suggest that households may still shy away from longer-term financial commitments, as is the case with home purchase. This finding aligns with Dutch data that show, for a period of seven years from 2009 to 2016, that positive economic expectations do not translate into plans to buy a home nor actual home purchase. It is in contrast to lesser expenses, such as car, appliance, and furniture purchases and vacations, where purchase intentions and actual purchases were shown to increase with more positive economic outlook [44]. In addition, our finding that financial stress and one-year economic expectations are not associated

with home equity balances in this cross-sectional study may point to the fact that time plays a critical role for these relationships. For example, longitudinal data have documented that, over time, positive economic expectations are associated with less conservative financial behaviors, such as holding mortgage debt rather than repaying the debt as shown in data from the British Household Panel Surveys [50] and with a greater likelihood of home equity borrowing as shown in panel data from the U.S. Survey of Consumer Expectations [45].

With regard to demographic predictors of homeownership, our study confirms long-established, well-documented associations [51], which speaks to the external validity of our findings. For example, regression results find an association of greater odds of homeownership with older age, married status, and having children. The odds of owning a home is inversely associated with Black race and lower educational attainment. The likelihood of higher home equity values is associated, as expected, with older age, higher education, children, self-employment, and retirement and inversely with Black race and lower educational attainment.

From a policy perspective, our findings suggest that housing-focused efforts in the military, such as VA loan products, relocation allowances, financial education and counseling programs, help military households cope with the demands of military career paths and the transition to post-active life. These requirements of military careers appear to not hamper the access to home ownership for current and former military service personnel more than the transitions and economic shocks experienced in civilian society (H1, H2). The assumption of economic shocks to the Life-Cycle Hypothesis of Savings are not supported by these results. A direction for future research is to collect data on a sufficiently large sample on the dates of home purchase among military households to disentangle whether military households were able to "catch up" or are in fact owning homes in similar rates as the civilian society across age groups.

When limiting the sample to homeowners (H3, H4), the data indicate lower housing wealth accumulation among the younger, Post-Vietnam era military households, compared to civilian households. As frequent military moves may prevent these households from building housing wealth while in the service, this group has had less time to accumulate housing wealth, confirming the role of housing tenure length for wealth accumulation also for this unique population group.

Potential directions for future research include the suggestion to examine at what point the younger, Post-Vietnam era military households catch-up with civilian households in their housing wealth accumulation. In addition, future research may break up the more diverse Post-Vietnam era cohort into several cohorts, for example breaking the sample again in 2001 with the start of the Global War on Terrorism era. This would avoid statistical effects that may cancel out themselves and allow for alignment with other ongoing research, such as the Department of Defense' Millenium Cohort Study [26].

Limitations to this study must be addressed. First, the cross-sectional nature of the Survey of Consumer Finances limits the analyses. Causality or directionality of the associations cannot be addressed or inferred. Second, the Survey of Consumer Finances does not ask about when a respondent retired from active-duty military service. For this reason, a closer examination of active-duty vs post duty housing wealth and the transition into civilian life cannot be done with these data. Third, the Survey of Consumer Finances does not have a question about when a respondent became a homeowner. For this reason, home ownership status and home equity accumulation cannot be delineated. Fourth, the small sample size of military service personnel within the Survey of Consumer Finances limited the robustness of the subsample analyses. A fully interacted specification might be preferrable. Finally, sample selection bias is a significant concern when analyzing military personnel, except for studies that control for this bias by focusing primarily on military members who were drafted into service.

## Supporting information

**S1 Text. Questionnaire text.**
(DOCX)

**S1 Table. Literature review of financial variables.**
(DOCX)

**S1 List. References for literature review.**
(DOCX)

**S2 Table. Regression of anxiety on financial stress.**
(DOCX)

**S3 Table. Sample characteristics by military status.**
(DOCX)

## Author contributions

**Conceptualization:** Eric Olsen, Cäzilia Loibl, Andrew Hanks.

**Data curation:** Eric Olsen, Andrew Hanks.

**Formal analysis:** Eric Olsen, Sherman D. Hanna, Andrew Hanks.

**Funding acquisition:** Eric Olsen.

**Investigation:** Eric Olsen.

**Methodology:** Eric Olsen, Cäzilia Loibl, Sherman D. Hanna, Andrew Hanks.

**Project administration:** Eric Olsen.

**Resources:** Eric Olsen, Cäzilia Loibl.

**Software:** Eric Olsen, Andrew Hanks.

**Supervision:** Cäzilia Loibl, Andrew Hanks.

**Validation:** Eric Olsen, Andrew Hanks.

**Visualization:** Eric Olsen.

**Writing – original draft:** Eric Olsen.

**Writing – review & editing:** Eric Olsen, Cäzilia Loibl, Sherman D. Hanna, Andrew Hanks.

## Acknowledgments

The authors would like to thank the FINRA Investor Education Foundation for providing access to the restricted National Financial Capability Study data.

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
