## [Decision Letter · Decision Letter 0]

4 Apr 2025

PONE-D-25-06259Differences in Housing Wealth between U.S. Military Service Personnel and the Civilian Population—Exploring the Role of Financial StressPLOS ONE

Dear Dr. Loibl,

Thank you for submitting your manuscript to PLOS ONE. After careful consideration, we feel that it has merit but does not fully meet PLOS ONE’s publication criteria as it currently stands. Therefore, we invite you to submit a revised version of the manuscript that addresses the points raised during the review process.

We look forward to receiving your revised manuscript.

Kind regards,

Zhou Yu, PhD

Academic Editor

PLOS ONE

Journal Requirements:

Additional Editor Comments:

Dear Authors,

I have completed the review. Based on the recommendations of the reviews, I decide to extend an opportunity to revise the manuscript. Here are some major concerns:

1. Context and Contribution: Strengthen the introduction by clearly outlining the research problem and the significance of findings.

2. Interpretation of Results: Expand on key findings, particularly the unexpected relationship between economic expectations and housing outcomes. Explain why lower economic expectations correlate with better housing outcomes.

3. Veteran Cohort Classification: Justify the categorization of military personnel into two broad cohorts, ensuring it accurately reflects differing military experiences and stressors. Have researchers used similar classification in the past? Further justify the choice of military personnel as the study population.

4. Financial Stress and Veteran Status: Address the inability to distinguish between active-duty personnel and veterans in the dataset and how this limitation impacts the conclusions. Are DoD civilian employees included in this analysis? Have you separately identified Reserve and National Guard Personnel?

5. Interpretation of Findings: "The odds of Korea/Vietnam Era military service personnel households owning a home is 4.23 times as large as the odds of civilian households owning a home (p<0.001).?" seems inconsistent with what is reported in the table.

6. Discussion Section: Enhance the discussion of statistically significant findings, particularly the role of demographic factors in housing disparities.

There are also minor concerns:

1. Clarity and Grammar: Revise the manuscript to correct grammatical errors and ensure sentence completion. For instance, please revise the sentence "The findings do not indicate that differences in financial stress between military and civilian households were associated with home equity accumulation." and make it more readable. Please revise "Vietnam Era military service personnel households had a positive but non-statistically significant relationship to home ownership?"

2. Introduction Structure: Revise the introduction for clarity and impact, ensuring the research question is presented explicitly.

3. Literature Review: More explicitly highlight the research gaps and how the study addresses them.

4. Methods Section: Consider relocating some data processing details to the appendix for improved readability.

5. Formatting Issues: Correct inconsistencies in table formatting, indentation, and citation styles.

6. Conclusion: Clearly articulate the study’s contributions, innovations, and future research directions. If this study has resolved conflicts in the literature, please state it clearly. Has the research discovered new things? The findings seem to reflect those of the whole population--being Black is associated with lower homeownership probabilities.

Reviewers' comments:

Reviewer's Responses to Questions

**Comments to the Author**

1. Is the manuscript technically sound, and do the data support the conclusions?

Reviewer #1: Yes

Reviewer #2: Yes

Reviewer #3: Yes

2. Has the statistical analysis been performed appropriately and rigorously? 

Reviewer #1: Yes

Reviewer #2: Yes

Reviewer #3: Yes

3. Have the authors made all data underlying the findings in their manuscript fully available?

Reviewer #1: Yes

Reviewer #2: Yes

Reviewer #3: Yes

4. Is the manuscript presented in an intelligible fashion and written in standard English?

Reviewer #1: Yes

Reviewer #2: No

Reviewer #3: No

5. Review Comments to the Author

Reviewer #1: This manuscript is methodologically sound, concise, and focused. It examines differences in home ownership and housing equity between veterans and civilians and draws from economic frameworks to present clear hypotheses with logical relationships between outcome and explanatory variables. The manuscript, however, lacks context specifying the nature of the problem the authors hope to contribute to and the importance of their findings. Given the methodological strength, I suggest a re-submission.

1. The study analyzes secondary data from multiple sources. Its dependent variables come from the Survey of Consumer Finances, while its explanatory measure of financial stress comes from the National Financial Capability Study. This choice removes common method bias and increases the study’s internal validity. Its other explanatory variable, veteran status, is an objective measure and is unlikely to introduce bias to the results. The authors’ justification for their models and their explanation of their baseline specification is clear and well-reasoned.

2. The authors find support for just one of their four hypotheses – post-Vietnam era military households were associated with lower levels of home equity than civilian households, which the authors argue likely relates to the frequency of geographical re-locations for military members. Also among the key findings is a consistent, perhaps peculiar relationship between economic expectations and homeownership.

a. Logistic regression models 2 and 3 show that believing the economy would become worse in the next year was associated with a significantly higher likelihood of owning a home, and believing the economy would improve was associated with a significantly lower likelihood of owning a home. This effect partly holds in OLS Model 2, as worse economic expectations were associated with greater home equity.

b. This finding is intriguing, but the manuscript’s discussion does not interpret its meaning or value. The overall interpretation of the findings is lacking and limits the manuscripts’ contribution. This omission leads me to suggest a second submission. The paper is methodologically sound, and I believe the authors can bolster their discussion section in a revision.

3. While the authors clearly state the value of their findings in contrasting theoretical expectations on the relationship between military status and home ownership, they would benefit from interpreting the results that did yield statistical significance. For example, it is unclear why, in the authors’ minds, lower economic expectations are associated with better housing outcomes or why demographic factors (e.g., race) were consistently associated with negative outcomes. The latter finding may be more obvious, but the manuscript would benefit from a more robust discussion. The study’s rigor is a notable strength, and its methods are presented well, but the lack of interpretation does not allow for a clear conceptual or empirical takeaway.

These major concerns are located in the paper’s back end. The following minor concerns may also merit consideration.

1. The research question on page 4 is presented (grammatically) as a statement. Additionally and more importantly, the statement of the problem on page 4 (“Due to the critical role of housing wealth”) lacks impact. A clearer introduction of the relevant problem and a clearer statement of the first research question would help set a stronger foundation for the methods and results.

2. Perhaps this is a reader issue more than a writer issue, but I did not quite understand section 1.3 of the introduction. The bulk of the preceding content presents evidence about negative financial outcomes for veterans, but section 1.3 ends by vaguely noting financial outcomes have improved. This improvement would seem to contradict Hypothesis 2.

3. The paper is well-written throughout, but like any manuscript, there are minor issues to clean up. I only note the instances below because they did distract from my first read of the manuscript.

a. The paragraph at the top of page 8 (continued from page 7) is missing a verb after “descriptive analyses.”

b. At least one in-text citation is presented in parenthesis rather than brackets. I understand this note is pedantic. I include it because it did briefly disrupt the way I read the manuscript. On first glance, the (47) appeared as some sort of specification or clarification rather than a citation.

c. The “Predictor – financial stress” paragraph in section 2.2 includes an unnecessary indent. Table 1, just below this section, is formatted noticeably differently than the other tables in the manuscript.

Reviewer #2: This article presents interesting and important information regarding service member homeownership and equity attainment. The data analysis is described in a way that makes it easy to follow and choices about which data to include or exclude were described satisfactorily. The topic of the paper is interesting and presented in a way that is easily digestible.

However, there are some concerns with the hypothesis and some of the data choices.

The authors split military service personnel into two distinct cohorts. Those who served during the Vietnam/Korean war era, and those who served in the post-Vietnam era. Anyone 18 or older in 1973 was included in the Vietnam era, anyone younger than 18 was included in the post-Vietnam era. This likely leads to at least some people who were military age in 1973 but did not join the military until after the war being included in the Vietnam era cohort.

Military experiences and stressors have changed greatly from the end of the Vietnam war to the current era. Those who have served post-2001 have often seen much higher levels of combat, deployments, and stress than those who served between the Vietnam war and the war on terror. The authors should discuss why the broad range of the second cohort is appropriate.

Another concern is that in the literature review the authors explain that active-duty military members often have more financial stress than their civilian counterparts, but that military veterans have been found to be more financially stable. Given that there is no way in the survey data to differentiate active duty personnel from veterans, or those who served for 4 years from those who retired with a pension for that matter, it may be that the lower performance of active duty personnel combined with the higher performance of veterans worked together to show that there is no difference in homeownership or equity between service personnel and the civilian population. The authors should explain better why they came to the hypotheses that they came to given the information in the literature review.

Finally, there were at least two instances where sentences seemed to cut off in the middle and were left incomplete. This may be due to something being cut accidentally during the editing process. There are also several grammar errors throughout the manuscript that affect the readability of the article. The authors should go through and fill in the missing information and comb the narrative for grammar errors to increase the clarity of the paper.

This is an important topic, and the results point to interesting avenues for further research. I commend the authors for their work and for the interesting direction of their inquiry. With a little cleanup, I think this article can be made much stronger.

Reviewer #3: 1 There are grammatical inaccuracies in both the abstract and the main text; a thorough revision of the language is required.

2 It is recommended to streamline the introduction to avoid excessive information and ensure clarity.

3 The literature review should explicitly highlight the existing research gap.

4 The Methods section appears somewhat lengthy; it is advisable to relocate part of the data processing details to the appendix to improve readability.

5 The conclusion should more explicitly articulate the study’s contributions and innovations; additionally, it is recommended to include a paragraph outlining potential directions for future research.

6. PLOS authors have the option to publish the peer review history of their article (what does this mean? ). If published, this will include your full peer review and any attached files.

**Do you want your identity to be public for this peer review?** For information about this choice, including consent withdrawal, please see our Privacy Policy .

Reviewer #1: No

Reviewer #2: **Yes: ** Robert Thomas Porter

Reviewer #3: No

---

## [Author Response · Author response to Decision Letter 1]

7 Aug 2025

EDITOR COMMENTS

EDITOR COMMENT 1. Context and Contribution: Strengthen the introduction by clearly outlining the research problem and the significance of findings.

AUTHORS’ RESPONSE: Your point is well taken. We completely revised the introduction section. The first subsection is completely revised and states the research questions. The next four subsections now explicitly state the research gap and our working hypotheses. In addition, we revised two of the four research hypotheses to more accurately reflect the current literature, thanks to Reviewer 2’s comments.

EDITOR COMMENT 2. Interpretation of Results: Expand on key findings, particularly the unexpected relationship between economic expectations and housing outcomes. Explain why lower economic expectations correlate with better housing outcomes.

AUTHORS’ RESPONSE: Your point is well taken. We added two new paragraphs that discuss these findings in the Discussion section. In addition, we expanded the literature review on this relationship in the introductory sections.

EDITOR COMMENT 3. Veteran Cohort Classification: Justify the categorization of military personnel into two broad cohorts, ensuring it accurately reflects differing military experiences and stressors. Have researchers used similar classification in the past? Further justify the choice of military personnel as the study population.

AUTHORS’ RESPONSE: Thank you for pointing out the need for further justification. Following this suggestion we added additional information about the larger post-Vietnam cohort. Unfortunately, due to sample size limitations – we have only about 300 responses in the post-Vietnam sample – we decided against dividing the subsample further. We added a statement about this limitation at the end of the section “Differences in military service personnel by enlistment and era of military service” and also point out the need for further research in this regard.

EDITOR COMMENT 4. Financial Stress and Veteran Status: Address the inability to distinguish between active-duty personnel and veterans in the dataset and how this limitation impacts the conclusions. Are DoD civilian employees included in this analysis? Have you separately identified Reserve and National Guard Personnel?

AUTHORS’ RESPONSE: Following your suggestion, the revised manuscript now acknowledges early on, in section “Measures,” the inability to distinguish between active-duty personnel and veterans in the dataset, stating, “We acknowledge that this question does not allow us to distinguish between active-duty and discharged or retired military personnel; this fact should be kept in mind when interpreting the results. Department of Defense civilian employees do not count as military personnel. We divide the military households, which include the national guard, into two cohorts …”

Department of Defense civilian employees are not counted as military personnel. The National Guard is included in the definition of current and former military personnel.

EDITOR COMMENT 5. Interpretation of Findings: "The odds of Korea/Vietnam Era military service personnel households owning a home is 4.23 times as large as the odds of civilian households owning a home (p<0.001).?" seems inconsistent with what is reported in the table.

AUTHORS’ RESPONSE: Based on this comment, the revised manuscript now states more clearly that the information in the table has been transferred into odds ratios for easier interpretation in the text narrative, stating for example on p. 19, “When converting the coefficients shown in Table 2 to into odds-ratios by taking the exponential value of the unrounded coefficient of Korea/Vietnam Era, we obtain an odds ratio. The odds of Korea/Vietnam Era military service personnel households owning a home is 4.23 times as large as the odds of civilian households owning a home (p<0.001).”

EDITOR COMMENT 6. Discussion Section: Enhance the discussion of statistically significant findings, particularly the role of demographic factors in housing disparities.

AUTHORS’ RESPONSE: Following your suggestion, we added the following paragraph to the Discussion section on p. 26/27 stating, “With regard to demographic predictors of homeownership, our study confirms long-established, well-documented associations [51], which speaks to the external validity of our findings. For example, regression results find an association of greater odds of homeownership with older age, married status, and having children. The odds of owning a home is inversely associated with Black race and lower educational attainment.”

EDITOR COMMENT 7. Clarity and Grammar: Revise the manuscript to correct grammatical errors and ensure sentence completion. For instance, please revise the sentence "The findings do not indicate that differences in financial stress between military and civilian households were associated with home equity accumulation." and make it more readable. Please revise "Vietnam Era military service personnel households had a positive but non-statistically significant relationship to home ownership?"

AUTHORS’ RESPONSE: Based on this comment, all authors made an effort to read the manuscript with regard to grammar and sentence completion. We also revised the abstract to provide more complete information and edited the sentence in the abstract you noted in this revision.

EDITOR COMMENT 8. Introduction Structure: Revise the introduction for clarity and impact, ensuring the research question is presented explicitly.

AUTHORS’ RESPONSE: Your point is well taken. We completely revised the Introduction section. We removed writing that was not directly related to the data analysis and added the research questions in question format at the end of the Introduction section. We also revised the statement of the problem. A new paragraph at the end of the Introduction section now states specifically on p. 4, “Specifically, this study asks, does the older, Korea/Vietnam era cohort of current and former military service personnel achieve housing wealth that is similar to civilian households? Does the younger, post-Vietnam era cohort of current and former military service personnel report lower housing wealth compared to civilian households? A particular focus in this context is on the link between financial stress and housing wealth, examining both individual financial stress and expectations about the economy.”

EDITOR COMMENT 9. Literature Review: More explicitly highlight the research gaps and how the study addresses them.

AUTHORS’ RESPONSE: Your point is well taken. At the end of each section of the literature review, the revised study now explicitly points out the research gap, such as in this statement at the end of section “Economic frameworks for housing wealth accumulation” on p. 5, “The transitory aspect of military careers presents economic shocks unique to service members. The Behavioral Life Cycle framework suggests that consequently these individuals may delay housing wealth accumulation. In contrast, the financial capability framework suggests that military-specific support services and financial products can support housing wealth accumulation at a rate equal to the civilian population. These opposing perspectives point to a gap in available research.”

EDITOR COMMENT 10. Methods Section: Consider relocating some data processing details to the appendix for improved readability.

AUTHORS’ RESPONSE: Following this comment, we moved the development of the financial stress measure to S2 Appendix, reducing the Methods section by more than one page.

EDITOR COMMENT 11. Formatting Issues: Correct inconsistencies in table formatting, indentation, and citation styles.

AUTHORS’ RESPONSE: We use EndNote for citations to keep references consistent and we checked all references for completeness. We carefully checked table formatting and indentation.

EDITOR COMMENT 12. Conclusion: Clearly articulate the study’s contributions, innovations, and future research directions. If this study has resolved conflicts in the literature, please state it clearly. Has the research discovered new things? The findings seem to reflect those of the whole population--being Black is associated with lower homeownership probabilities.

AUTHORS’ RESPONSE: Thank you for pointing out these omissions. The revised Discussion section now states explicitly the study contributions and innovations and we also added a paragraph outlining potential directions for future research on p. 27/28, per Reviewer 3, stating, “Potential directions for future research include the suggestion to examine at what point the younger, Post-Vietnam era military households catch-up with civilian households in their housing wealth accumulation. In addition, future research may break up the more diverse Post-Vietnam era cohort into several cohorts, for example breaking the sample again in 2001 with the start of the Global War on Terrorism era. This would avoid statistical effects that may cancel out themselves and allow for alignment with other ongoing research, such as the Department of Defense’ Millenium Cohort Study [26].”

REVIEWER COMMENTS

REVIEWER 1

This manuscript is methodologically sound, concise, and focused. It examines differences in home ownership and housing equity between veterans and civilians and draws from economic frameworks to present clear hypotheses with logical relationships between outcome and explanatory variables. The manuscript, however, lacks context specifying the nature of the problem the authors hope to contribute to and the importance of their findings. Given the methodological strength, I suggest a re-submission. The study analyzes secondary data from multiple sources. Its dependent variables come from the Survey of Consumer Finances, while its explanatory measure of financial stress comes from the National Financial Capability Study. This choice removes common method bias and increases the study’s internal validity. Its other explanatory variable, veteran status, is an objective measure and is unlikely to introduce bias to the results. The authors’ justification for their models and their explanation of their baseline specification is clear and well-reasoned.

AUTHORS’ RESPONSE: Thank you!

REVIEWER 1 COMMENT 1. The authors find support for just one of their four hypotheses – post-Vietnam era military households were associated with lower levels of home equity than civilian households, which the authors argue likely relates to the frequency of geographical re-locations for military members. Also among the key findings is a consistent, perhaps peculiar relationship between economic expectations and homeownership.

Logistic regression models 2 and 3 show that believing the economy would become worse in the next year was associated with a significantly higher likelihood of owning a home, and believing the economy would improve was associated with a significantly lower likelihood of owning a home. This effect partly holds in OLS Model 2, as worse economic expectations were associated with greater home equity. This finding is intriguing, but the manuscript’s discussion does not interpret its meaning or value. The overall interpretation of the findings is lacking and limits the manuscripts’ contribution. This omission leads me to suggest a second submission. The paper is methodologically sound, and I believe the authors can bolster their discussion section in a revision.

While the authors clearly state the value of their findings in contrasting theoretical expectations on the relationship between military status and home ownership, they would benefit from interpreting the results that did yield statistical significance. For example, it is unclear why, in the authors’ minds, lower economic expectations are associated with better housing outcomes or why demographic factors (e.g., race) were consistently associated with negative outcomes. The latter finding may be more obvious, but the manuscript would benefit from a more robust discussion. The study’s rigor is a notable strength, and its methods are presented well, but the lack of interpretation does not allow for a clear conceptual or empirical takeaway. These major concerns are located in the paper’s back end.

AUTHORS’ RESPONSE: Your point is well taken. We made several and extensive adjustments following your comment. First, we added additional background information about the role of future expectations for household financial behaviors to the introductory section “Association of financial stress and housing wealth”, stating on p. 10/11 that, “In addition to financial stress experienced by households, expectations about the larger economic situation have been linked to housing wealth, pointing to the need for two types of stress measures. Individuals with more negative expectations about the greater economy have been shown to be more conservative in their consumption, credit, and investment behavior, while the opposite is true for individuals with optimistic expectations [44]. With regard to home equity, more uncertain expectations about the economy have been linked to lower likelihood of investing in risky assets, pointing to potential preferences for the stability of housing wealth, and to household behavior that shows more precaution in borrowing against the equity in the home, for example, with home equity lines of credit [45, 46]. Our working hypothesis in response to this gap in research is that financial stress is associated with lower levels of home ownership and home equity. In contrast, we hypothesize that more negative expectations about the economy are associated with conservative financial behaviors, including a greater interest in investing in home purchase and home equity growth.”

Second, in the discussion section, we significantly expanded the interpretation of the regression results, adding two new paragraphs on p. 25/26. These new paragraphs specifically address the relationship of economic expectations and housing wealth, stating, “The study further documents the role of individual financial stress, economic expectations, and demographic factors in the context of housing wealth. Greater individuals financial stress is found to be associated with lower homeownership rates, documenting that financial stress has multiple facets, as captured in this newly developed composite measure and that they relate to the home buying decision. The individual financial stress measure introduced in this study is derived from debt-to-income ratio, payday loan use, missed payments, credit card balance, lack of emergency savings, financial risk tolerance, and financial knowledge.

The results for economic expectations indicate that individuals with better one-year expectations are less likely to own a home compared to those that do not expect changes in the economy. This finding does not align with the initial working hypothesis that a more positive economic outlook is associated with homeownership, as compared to renting. The literature on association of economic expectations and household economic behavior documents that more positive economic expectations tend to make households take greater financial risks, such as through new borrowing and credit line increases [46]. Based on one-year frames of economic expectations, our findings suggest that households may still shy away from longer-term financial commitments, as is the case with home purchase. This finding aligns with Dutch data that show, for a period of seven years from 2009 to 2016, that positive economic expectations do not translate into plans to buy a home nor actual home purchase. It is in contrast to lesser expenses, such as car, appliance, and furniture purchases and vacations, where purchase intentions and actual purchases were shown to increase with more positive economic outlook [44]. In addition, our finding that financial stress and one-year economic expectations are not associated with home equity balances in this cross-sectional study may point to the fact that time plays a critical role for these relationships. For example, longitudinal data have documented that, over time, positive economic expectations are associated with less conservative financial behaviors, such as holding mortgag

---

## [Editor Report · Decision Letter 1]

15 Aug 2025

Differences in Housing Wealth between U.S. Military Service Personnel and the Civilian Population—Exploring the Role of Financial Stress

PONE-D-25-06259R1

Dear Dr. Loibl,

We’re pleased to inform you that your manuscript has been judged scientifically suitable for publication and will be formally accepted for publication once it meets all outstanding technical requirements.

Kind regards,

Zhou Yu, PhD

Academic Editor

PLOS ONE
---

## [Editor Report · Acceptance letter]

PONE-D-25-06259R1

PLOS ONE

Dear Dr. Loibl,

I'm pleased to inform you that your manuscript has been deemed suitable for publication in PLOS ONE. Congratulations! Your manuscript is now being handed over to our production team.

Kind regards,

on behalf of

Dr. Zhou Yu

Academic Editor

PLOS ONE